# Towards Semantic Equivalence of Tokenization in Multimodal LLM

**Shengqiong Wu**[1*], **Hao Fei**[1†], **Xiangtai Li**[2], **Jiayi Ji**[1], **Hanwang Zhang**[3],
**Tat-Seng Chua**[1], **Shuicheng YAN**[4,1]
[1]National University of Singapore, [2]ByteDance Seed,
[3]Nanyang Technological University, [4]Skywork AI
`swu@u.nus.edu`,`{haofei37, yansc, dcscts}@nus.edu.sg`,
`{jjyxmu, xiangtai94}@gmail.com`,`hanwangzhang@ntu.edu.sg`

## Abstract

Multimodal Large Language Models (MLLMs) have demonstrated exceptional capabilities in processing vision-language tasks. One of the crux of MLLMs lies in vision tokenization, which involves efficiently transforming input visual signals into feature representations that are most beneficial for LLMs. However, existing vision tokenizers, essential for semantic alignment between vision and language, remain problematic. Existing methods aggressively fragment visual input, corrupting the visual semantic integrity. To address this, this work presents a novel dynamic Semantic-Equivalent Vision Tokenizer (**SeTok**), which groups visual features into semantic units via a dynamic clustering algorithm, flexibly determining the number of tokens based on image complexity. The resulting vision tokens effectively preserve semantic integrity and capture both low-frequency and high-frequency visual features. The proposed MLLM (**Setokim**) equipped with SeTok significantly demonstrates superior performance across various tasks, as evidenced by our experimental results. The project page is `https://sqwu.top/SeTok-web/`.

## 1 Introduction

Recently, the research on MLLMs has garnered intense interest (Zhang et al., 2024a; Lin et al., 2023; Dong et al., 2024; Wu et al., 2024a). By building upon the unprecedented intelligence of language-based LLMs (Chiang et al., 2023; Touvron et al., 2023a), and integrating multimodal encoders (Radford et al., 2021) at the input side and decoders (Rombach et al., 2022) at the output side, current MLLMs have developed powerful multimodal capabilities. Particularly, in the visual modality, the state-of-the-art (SoTA) MLLMs have now achieved a grand slam across the four major visual-language task groups, i.e., understanding (Liu et al., 2024b; Wu et al., 2024c; Team, 2024), generating (Ge et al., 2023; Dong et al., 2024; Jin et al., 2024b; Pan et al., 2024), segmenting (Ren et al., 2024; You et al., 2023), and editing (Huang et al., 2024; Jin et al., 2024b; Fu et al., 2024). Central to this capability is the design of vision tokenization (Dosovitskiy et al., 2021; Esser et al., 2021; Yu et al., 2024), which focuses on effectively converting input visual signals into visual tokens that can be seamlessly understood by LLMs. Existing vision tokenizers primarily produce three types of visual tokens: 1) patch-level continuous tokens (cf. Figure 1(a)), 2) patch-level discrete tokens (cf. Figure 1(b)), and 3) learnable query tokens (cf. Figure 1(c)).

While existing MLLMs have achieved promising performances across various tasks, a significant bottleneck remains with current visual tokenization methods, i.e., resulting in insufficient semantic alignments between language and vision tokens. On the language side, linguistic tokens (or words) are naturally discrete, representing well-encapsulated semantic units, whereas, on the vision side, visual pixels are inherently continuous data with no physical boundaries. Intuitively, language tokens should correspond to semantically encapsulated objects (or compositional regions) within an image. For example, when "*a dog*" is mentioned, the "*dog*" token should correspond to the direct pixel

---

*Work done when Shengqiong Wu is an intern at Skywork AI.
†Corresponding Author.

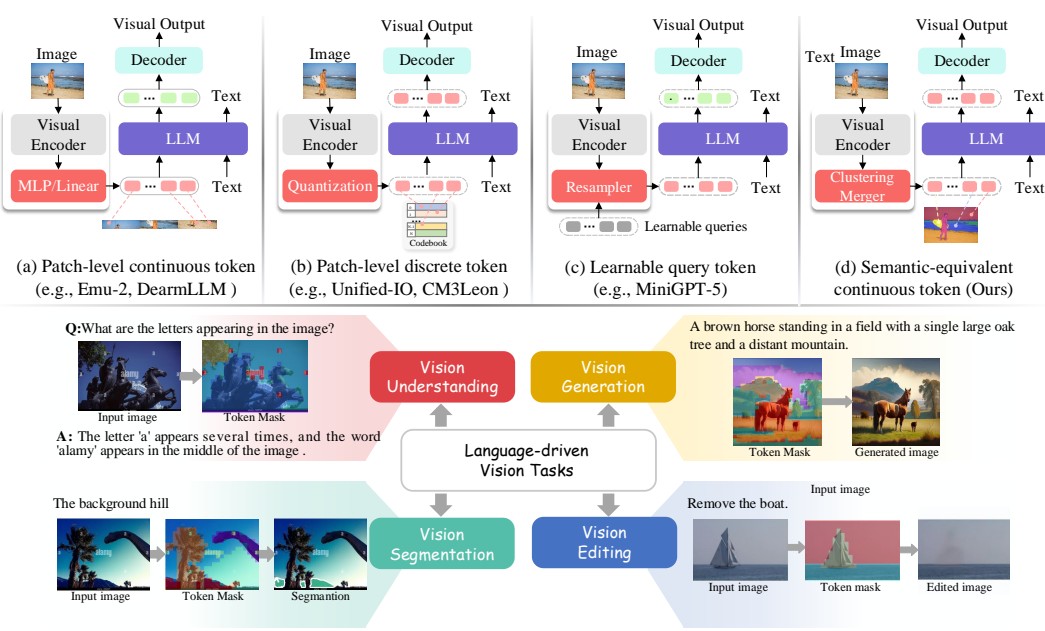

Figure 1: Comparison between existing MLLMs in tokenized visual inputs: (a) patch-level continuous token, (b) patch-level discrete token, (c) learnable query token, and (d) semantic-equivalent continuous token (ours). In (e), we show four language-driven vision tasks enhanced with semantic-equivalent vision tokens, with token masks showing regions of the same color representing a single vision token.

region of the dog in the image. However, as illustrated in Figure 1(a&b), both existing tokenization methods divide the image into fixed patch squares, fragmenting objects across multiple patches. This disrupts the integrity of visual semantic units, resulting in a significant loss of high-frequency visual information (Zhang et al., 2023), e.g., the object's edges and contours. Moreover, methods employing a fixed number of query tokens, as depicted in Figure 1(c), struggle to align with actual visual semantic units and meanwhile offer limited interpretability (Yang et al., 2022; Wu et al., 2024b). Ultimately, this misalignment between vision and language within MLLMs undermines the effective understanding of visual signals, significantly hindering progress in a range of vision-language tasks that require precise, fine-grained semantic alignment between vision and language elements.

To this end, this work proposes a Semantic-Equivalent Tokenizer (**SeTok**) for enhancing MLLMs, where we encourage the vision and language tokens to be semantically congruent. The core idea involves automatically grouping visual features from input images by applying a clustering algorithm (Engelcke et al., 2021), such that each unique cluster represents an encapsulated semantic unit within the vision. As illustrated in Figure 1(d), the red visual area aggregated by SeTok corresponds to a complete semantic concept—"*person*", while the yellow area corresponds to the "*surface board*" concept. Furthermore, we recognize that tokenizing images into a fixed number of patches is impractical. From a semantic perspective, different images should contain varying numbers of semantically encapsulated objects, and the granularity of compositional regions also needs to be flexibly determined. For example, we only need to identify a person in the image, while at other times, we may need to delineate the person's head precisely. This implies that it is more reasonable to dynamically determine the division of visual tokenization. To address this, we propose a dynamic clustering mechanism (Engelcke et al., 2021) that iteratively determines cluster centers based on density peaks, assigning visual features to these centers until all features are allocated. The design of this mechanism allows for the dynamic determination of the number of concept visual tokens, rather than fixing the ratio (Jin et al., 2024a) or merely merging the top-$k$ visual tokens (Bolya et al., 2023). After clustering, we devise a token merger to aggregate the visual features within each cluster, that is dedicated to learning a complete visual semantic unit feature, including both high-frequency and low-frequency information. To enable the effective learning of the semantic-equivalent token, we propose reconstructing the raw image based on these tokens, and further introducing the concept-level image-text contrastive loss to explicitly align the language and vision at the concept level.

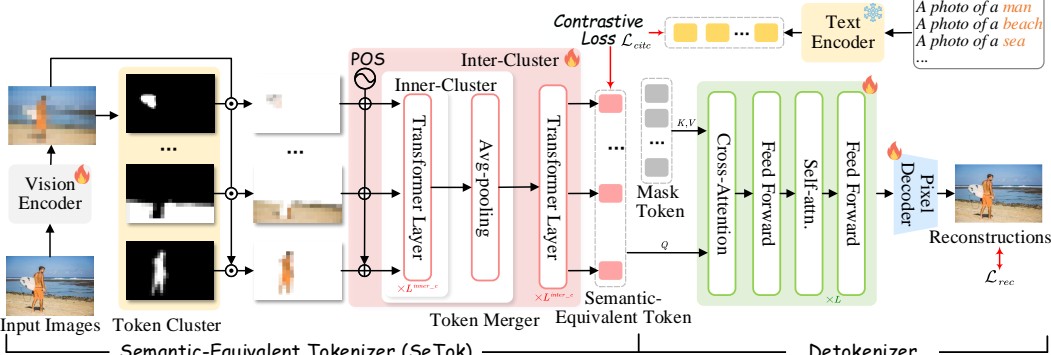

Figure 2: Overview of **SeTok**. **SeTok** tokenizes visual features extracted from an image by a vision encoder into semantically equivalent vision tokens, which then are fed into a detokenizer to reconstruct the image and meanwhile employed to perform the concept-level image-text alignment.

Built on a pre-trained LLM (Touvron et al., 2023b), **SETOKIM** performs reasoning on a unified multimodal sequence concatenating text token with visual tokens generated by **Setok**. During inference, **SETOKIM** yields the text and visual tokens autoregressively, which are then processed by a visual detokenizer and mask decoder to produce images and corresponding masks. Inspired by Li et al. (2024a), we introduce a unified autoregressive training objective for optimization through pre-training and instruction-tuning on the massive multimodal data. We evaluate **SETOKIM** on various common visual language tasks, including visual question-answering, image generation & editing, and referring segmentation. Our results reveal that semantic-equivalent tokenization significantly enhances vision-language learning compared to standard patch-level tokenization or learnable queries, achieving higher performance on various tasks. Meanwhile, in-depth analyses and visualizations intuitively show **SeTok** enjoys the superiority of tokenizing vision input into more interpretable and semantic-complete vision tokens, achieving fine-grained vision-language alignment.

## 2 METHODOLOGY

In this work, we aim to generate semantically complete vision tokens aligned with text tokens to facilitate fine-grained semantic interactions between vision and language, thereby enhancing the performance of MLLMs in various multimodal tasks. In pursuit of this goal, we propose constructing a semantic-equivalent tokenizer, called **SeTok**, which tokenizes the given input image into a sequence of semantically complete visual tokens, as illustrated in Figure 2. Integrated with the SeTok, we further design a multimodal large language model, i.e., **SETOKIM** shown in Figure 3, where the semantic-equivalent vision tokens concatenated with text tokens are fed into LLMs for interleaved image-text understanding and generation.

### 2.1 SEMANTIC-EQUIVALENT VISION TOKENIZER

Given an input image $I \in \mathbb{R}^{H \times W \times 3}$, we first employ a vision encoder to extract a sequence of visual patch embeddings $\boldsymbol{X} = \{\boldsymbol{x}_{i,j}\} \in \mathbb{R}^{h \times w \times d}$, where $d$ is the embedding dimension [1]. Then, to obtain semantically complete visual tokens, we propose to amalgamate the visual embeddings into concept-like scene components by a Token Cluster.

**Token Cluster.** We take the visual patch embeddings $\boldsymbol{X}$ as input and then assign individual patches into a semantic complete cluster, which can be formulated as obtaining a variable number of concept masks $\boldsymbol{M} \in [0, 1]^{h \times w \times C}$, with $\sum_c \boldsymbol{M}_{i,j,c} = 1$ for all patch coordinate tuples $(i, j)$ in an image, where $C$ is the number of semantic-equivalent tokens. Inspired by Engelcke et al. (2021), this is intuitively achieved by (1) selecting the location $(i, j)$ of the visual patch feature that has not yet been assigned to a cluster, (2) creating a cluster assignment according to the distance of the embeddings at the selected location to all other embeddings according to a distance kernel $\varphi(\cdot)$ [2], and (3) repeating the first two steps until all visual embeddings are accounted for or a stopping criterion is met. Different from (Du et al., 2016) employing uniformed seed scores performing the stochastic selection of visual embeddings, we propose to choose the visual embeddings based on their density

---

[1]When using ViT-based vision encoder, $h = \frac{H}{p}, w = \frac{W}{p}$, where $p$ is the patch size. Similarly, $p$ denotes the downsampling factor when using a CNN-based encoder.

[2]In this work, we employ $\varphi(\boldsymbol{u}, \boldsymbol{v}) = \exp(-\|\boldsymbol{u} - \boldsymbol{v}\|^2 \cdot C \ln 2)$

peaks, as a higher density shows a higher potential to be the cluster center. Specifically, we first calculate the local density $\rho_{i,j}$ of the token $\boldsymbol{x}_{i,j} \in \boldsymbol{X}$ by referring its neighbors:

$$\rho_{i,j} = \exp(-\frac{1}{K} \sum_{\boldsymbol{x}_{m,n} \in \text{KNN}(\boldsymbol{x}_{i,j}, \boldsymbol{X})} \varphi(\boldsymbol{x}_{m,n}, \boldsymbol{x}_{i,j})), \quad (1)$$

where $\text{KNN}(\boldsymbol{x}_{i,j}, \boldsymbol{X})$ denotes the $K$-nearest neighbors of $\boldsymbol{x}_{i,j}$ in $\boldsymbol{X}$. We then measure the minimal distance $\delta_{i,j}$ between the feature $\boldsymbol{x}_{i,j}$ and other features with higher density:

$$\delta_{i,j} = \begin{cases} \min_{m,n:\rho_{m,n}>\rho_{i,j}} \varphi(\boldsymbol{x}_{m,n}, \boldsymbol{x}_{i,j}), & \text{if } \exists\, m, n : \rho_{m,n} > \rho_{i,j} \\ \max_{m,n} \varphi(\boldsymbol{x}_{m,n}, \boldsymbol{x}_{i,j}), & \text{otherwise} \end{cases} \quad (2)$$

Finally, we summarize the score $s_{i,j}$ of the feature by combining the local density $\rho_{i,j}$ and minimal distance $\delta_{i,j}$ as $\rho_{i,j} \times \delta_{i,j}$. Based on the score, we select the location $(i, j)$ of the visual feature that has the highest score $s_{i,j}$ and has not yet been assigned to a cluster and iteratively assign the visual feature into a certain cluster until a stopping condition is satisfied, at which point the additional mask is added for any remaining visual embeddings. The detailed algorithm is described in Appendix §C.1.

**Token Merger.** After clustering, the visual embeddings are grouped based on the attention masks $\boldsymbol{M}$. To optimally retain information within each cluster, we adopt a token merger that aggregates visual embeddings beyond merely using cluster centers as definitive vision tokens. In addition, considering the significance of positional information for representing a semantic concept in an image, we integrate 2D position embeddings (PE, Heo et al. (2024)) into the merger, calculated as $\hat{\boldsymbol{X}}_c = \text{PE}(\boldsymbol{X}) \odot \boldsymbol{M}_c \oplus \boldsymbol{X} \odot \boldsymbol{M}_c$. Then, we apply $L^{\text{inner\_c}}$ Transformer layers on all the visual embeddings within a cluster, followed by an average pooling to obtain the final token feature $\boldsymbol{u_c} = \text{Avg}(\text{Transformer}(\hat{\boldsymbol{X}}_c), L^{\text{inner\_c}}) \in \mathbb{R}^d$. To facilitate the representation of coherent scenes with semantic equivalent tokens, we add inter-cluster Transformer layers to model relationships between vision tokens, i.e., $\boldsymbol{V} = \{\boldsymbol{v}_1, \cdots, \boldsymbol{v}_C\} = \text{Transformer}(\{\boldsymbol{u}_1, \cdots, \boldsymbol{u}_C\}, L^{\text{inter\_c}}) \in \mathbb{R}^{C \times d}$.

**SeTok Training.** To facilitate diverse visual understanding and generation tasks when building MLLMs, we argue that effective semantic-equivalent tokens should embody two key attributes: complete and enriched high-level semantic information, and undistorted pixel-level details. Therefore, we propose to include concept-level image-text contrastive loss and image reconstruction loss, as shown in Figure 2. During the training phase, to ensure each token's semantic independence and completeness, we adopt a concept-level image-text conservative loss, inspired by Xu et al. (2022). This loss aligns visual tokens with corresponding textual concepts semantically, thereby enhancing their suitability for integration in LLMs. Additionally, to ensure the tokens retain adequate pixel-level details, we feed these tokens into a detokenizer (Yu et al., 2024) to reconstruct the original image and calculate the reconstruction loss. Finally, we employ a weighted sum to combine the contrastive loss and reconstruction loss, optimizing both semantic fidelity and visual detail retention:

$$\mathcal{L}_{setok} = \alpha\mathcal{L}_{rec} + \beta\mathcal{L}_{citc}. \quad (3)$$

In practice, $\alpha$ and $\beta$ are set to 1. We use ImageNet-1K (Deng et al., 2009) for reconstruction learning and OpenImages (Kuznetsova et al., 2020) for both reconstruction and alignment learning.

## 2.2 SETOKIM

Upon acquiring **SeTok**, we propose to integrate it with the pre-trained LLM to construct an MLLM, i.e., **SE-TOKIM**. The overall framework is depicted in Figure 3. The input image will be tokenized into a sequence of semantic-equivalent visual tokens by **SeTok**, which are then concatenated with text tokens to form a unified multimodal sequence. To effectively distinguish between modalities and facilitate visual content generation, two special tokens, '`[Img]`' and '`[/Img]`' are introduced to signify the beginning and the end of the visual sequence, respectively. The backbone LLM subsequently processes this multimodal sequence to

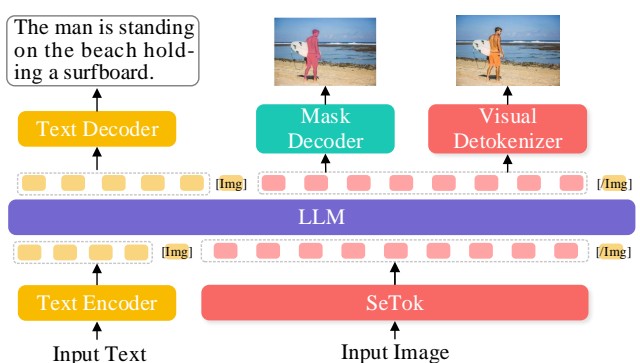

Figure 3: The overview of **SETOKIM**.

| Method | Size | Vis. Tok. | Flickr30K | VQA$^{v2}$ | OK-VQA | GQA | POPE | MME | MM-Vet |
|--------|------|-----------|-----------|------------|--------|-----|------|-----|--------|
| InstructBLIP (Liu et al., 2023a) | 13B | Q.C | - | - | - | 49.5 | 78.9 | 1212.8 | - |
| Qwen-VL-Chat (Bai et al., 2023) | 7B | Q.C | - | 78.2* | - | 57.5* | - | 1487.5 | - |
| Emu (Sun et al., 2024b) | 7B | Q.C | 77.4 | 57.2 | 43.4 | - | - | - | - |
| DreamLLM (Dong et al., 2024) | 7B | P.C | - | 72.9 | - | 41.8 | - | - | 36.6 |
| LLaVA-1.5 (Liu et al., 2024b) | 7B | P.C | - | 78.5* | - | 62.0* | 85.9 | 1510.7 | 33.1 |
| NExT-GPT (Wu et al., 2024c) | 7B | P.C | 84.5 | 66.7 | 52.1 | - | - | - | - |
| SEED-X (Ge et al., 2024) | 17B | P.C | 52.3 | - | - | 47.9 | 84.2 | 1435.7 | - |
| LaVIT (Jin et al., 2024b) | 7B | P.C | 83.0 | 66.0 | 54.6 | 46.8 | - | - | - |
| Unified-IO-2 (Lu et al., 2024) | 6.8B | P.D | - | **79.4*** | - | - | 87.7 | - | - |
| CM3Leon (Yu et al., 2023a) | 7B | P.D | - | 47.6 | 23.8 | - | - | - | - |
| Chameleon (Team, 2024) | 34B | P.D | 74.7 | 66.0 | - | - | - | - | - |
| **SETOKIM** | 7B | SE.C | **86.9** | 78.5* | **60.2*** | **65.6*** | **89.1** | **1537.8** | **45.2** |

Table 1: Comparison of MLLMs on image understanding benchmarks. * indicates the training sets observed during training. "C" and "D" represent continuous and discrete visual tokens, respectively. "P" refers to patch-level features, "Q" denotes learnable queries, and "SE" is semantic-equivalent.

perform multimodal understanding and generation. The output vision tokens are then fed into the visual detokenizer to restore the images. Meanwhile, we observe that the generated concept-centric tokens inherently embed approximate locations of each concept within the original image, as illustrated in 6. To exploit this spatial and semantic encoding, we incorporate a lightweight mask decoder (Kirillov et al., 2023) utilizing the generated vision tokens as input to yield the referring mask. The detailed implementations are provided in the Appendix §D.

**Training Objectives.** To facilitate autoregressive modeling across both text and visual generation, we unified adopt a next-token prediction:

$$p(y_1, \cdots, y_n) = \prod_{i=1}^{n} p(y_i | y_1, \cdots, y_i). \tag{4}$$

Specifically, in terms of text generation, we adopt the Cross-entropy loss $\mathcal{L}_{text}$, i.e., the standard language modeling objective, to maximize the likelihood of text tokens. For image generation, drawing inspiration from Li et al. (2024a), we utilize the LLM, to produce a conditioning vector $z_i$ based on previous tokens: $z_i = \text{LLM}(y_1, \cdots, y_{i-1})$. We then model the probability of the next token by $p(y_i | z_i)$, and employ the Diffusion loss (Li et al., 2024a), denoted $\mathcal{L}_{vis}$, to optimize the parameters of LLM. Moreover, we follow Li et al. (2024c) to use binary cross-entropy loss $\mathcal{L}_{bce}$ and dice loss $\mathcal{L}_{dice}$ for optimizing the parameters in mask decoder.

**Training Receipts.** Given that we try to perform generation and understanding with one single generative model, large-scale pretraining is required to achieve effective alignment between textual and visual content. To this end, we propose a two-stage training procedure. **Stage-I: Multimodal Pretraining.** In this stage, we focus on enhancing the alignment between text and image. We employ massive multimodal data, including ImageNet-1K and 28M text-image pair dataset, to train our model for conditional image generation and image captioning. Furthermore, we utilize the English text corpus from the SlimPajama (Soboleva et al., 2023) dataset to reduce catastrophic forgetting of the reasoning capacity during LLM training. Toward the end of the first phase of training, once the trainable modules of **SETOKIM** have converged, we freeze these modules and exclusively train the mask decoder on the segmentation datasets, like MSCOCO (Lin et al., 2014), to promote the learning of fine-grained object boundaries. **Stage-II: Instruction Tuning.** Building upon the pretrained weights, we further perform multimodal instruction tuning with both public datasets covering multimodal instruction datasets (e.g., ALLaVA (Chen et al., 2024) and LLaVA-665K (Liu et al., 2023c) ), fine-grained visual QA (e.g., VQA$^{v2}$ (Goyal et al., 2019), GQA (Hudson & Manning, 2019), OK-VQA (Marino et al., 2019), A-OKVQA (Schwenk et al., 2022)), image generation (e.g., LAION-aesthetics (Schuhmann et al., 2022)), and editing (e.g., InstructPix2Pix (Brooks et al., 2023) and Magicbrush (Zhang et al., 2024c)). More details can be found in the Appendix §D.3.

## 3 SETTINGS

Our experiments employ the LLaMA-2-7B (Touvron et al., 2023b) to initialize our LLM backbone. For the **SeTok**, we apply pre-trained SigLIP-SO400M-patch14-384 (Zhai et al., 2023) as our vision encoder, and the numbers of inner-cluster and inter-cluster transformer layers are set as 12, and 8, respectively. The dimension of the semantic-equivalent token is 512. For the detokenizer, we adopt $L = 12$ blocks for mask tokens to query visual information contained in the semantic-equivalent

| Method | Type | Decoder | MS-COCO | MagicBrush | | MA5K | | EVR | |
|---|---|---|---|---|---|---|---|---|---|
| | | | FID ↓ | CLIP$_{im}$ ↑ | L$_1$ ↓ | LPIPS↓ | L$_1$ ↓ | CLIP$_{im}$ ↑ | L$_1$ ↓ |
| ● **Diffusion-based Method** | | | | | | | | | |
| Make-A-Scene (Gafni et al., 2022) | Autoregressive | - | 11.8 | - | - | - | - | - | - |
| Ins.P2P* (Brooks et al., 2023) | Diffusion | - | - | 83.4 | 12.1 | 35.9 | 17.6 | 81.4 | 18.9 |
| SD v2.1 (Rombach et al., 2022) | Diffusion | - | 9.0 | - | - | - | - | - | - |
| ● **MLLM-based Method** | | | | | | | | | |
| DreamLLM (Dong et al., 2024) | LLM (Cont.) | SDv2.1 | 8.7 | - | - | - | - | - | - |
| SEED-X (Ge et al., 2023) | LLM (Cont.) | SDXL | 14.9 | - | - | - | - | - | - |
| CM3Leon (Yu et al., 2023a) | LLM (Disc.) | VQGAN | 10.3 | - | - | - | - | - | - |
| LaVIT* (Jin et al., 2024b) | LLM (Disc.) | SDv1.5 | **7.4** | 81.1 | 25.3 | 36.9 | 25.1 | 73.8 | 26.8 |
| LWM (Liu et al., 2024a) | LLM (Disc.) | VQGAN | 12.6 | - | - | - | - | - | - |
| MGIE* (Fu et al., 2024) | LLM (Cont.) | SDv1.5 | - | **91.1** | 8.2 | 29.8 | **13.3** | 81.7 | 16.3 |
| Emu-2-gen* (Sun et al., 2024a) | LLM (Cont.) | SDXL | - | 85.7 | 19.9 | 28.4 | 20.5 | 80.3 | 22.8 |
| Morph-Token* (Pan et al., 2024) | LLM (Cont.) | VQGAN | - | 87.9 | 7.6 | 27.9 | 14.6 | 82.6 | 15.3 |
| SETOKIM | LLM (Cont.) | SeTok | 8.5 | 89.6 | **6.9** | **26.4** | 15.7 | **83.5** | **14.1** |

Table 2: Performance of various models in zero-shot text-to-image generation and editing on benchmarks. *: denotes editing performances sourced from (Pan et al., 2024). "LLM (Cont.)" means LLM outputs continuous representation utilized in the decoder to generate images, while "LLM (Disc.)" stands for discrete representation generated for image generation.

tokens, and then an upsampler inspired by the architect of Yu et al. (2024) is employed as the pixel decoder. More implementation details are provided in Appendix §D.1.

For examining visual understanding ability, we evaluate our model on Flicker30K (Young et al., 2014), VQA$^{v2}$(Goyal et al., 2019), GQA (Hudson & Manning, 2019), OK-VQA (Marino et al., 2019), as well as three MLLM benchmarks, e.g., POPE (Li et al., 2023), MME (Fu et al., 2023) and MM-Vet (Yu et al., 2023b). Besides, we evaluate the visual generation fidelity on the MSCOCO (Lin et al., 2014) dataset. Following Pan et al. (2024), we evaluate the image editing capabilities of the SEKTOIM on Magicbrush (Zhang et al., 2024c), EVR (Tan et al., 2019) and MA5K (Shi et al., 2021). Furthermore, refCOCOg (Mao et al., 2016), refCOCO+ (Yu et al., 2016), and Reaseg (Lai et al., 2024) are utilized to examine the potential referring segmentation capabilities of the proposed model.

## 4 EXPERIMENTAL RESULTS

### 4.1 MAIN RESULTS

**The Quality of SeTok** We employ reconstruction FID (rFID) and Top-1 accuracy for image classification on ImageNet to measure the reconstruction and text alignment capabilities of the SeTok in Table 3. SeTok can achieve a comparable reconstruction quality to well-trained VQ models. Unlike prior methods that typically utilize 2D latent grids preserving spatial mappings between latent tokens and image patches, which al-

| Model | #Tokens | Latent size | rFID ↓ | Top-1 ↑ |
|---|---|---|---|---|
| VQ-GAN (Esser et al., 2021) | Fixed | $16 \times 16$ | 7.94 | - |
| VAE (Rombach et al., 2022) | Fixed | $32 \times 32$ | 2.63 | - |
| RQ-VAE (Lee et al., 2022) | Fixed | $16 \times 16$ | 3.20 | - |
| ViT-VQGAN (Yu et al., 2022) | Fixed | $32 \times 32$ | **1.28** | - |
| MQ-VAE (Huang et al., 2023) | Fixed | $32 \times 32$ | 5.29 | - |
| TiTok (Yu et al., 2024) | Fixed | $32 \times 1$ | 2.21 | 72.6 |
| **SeTok** | Dynamic | - | 2.07 | **75.4** |

Table 3: Reconstruction results (rFID) and image classification performance (Top-1 Accuracy) on $256 \times 256$ ImageNet(val.) dataset. #Tokens refers to the number of tokens.

lows for the retention of precise low-level information but limits high-level semantic acquisition and development of more compressed latent space, SeTok integrates both high- and low-level information that is crucial for producing high-quality images and creating semantic compact and complete latent representations. In comparison, the latest models like TiTok utilize a fixed number of 1D latent representations that suffer from a lack of semantic interpretability and poor textual alignment, i.e., obtaining inferior image classification performance (72.6 vs 75.4 top-1 accuracy). We visualize the visual token in Section 4.2, and more reconstruction examples can be found in Appendix §E.

**Visual Understanding.** We evaluate the visual understanding capabilities of our model and other leading MLLMs across a wide range of benchmarks, as detailed in Table 1. Different from the prevalent use of patch-level continuous visual tokens by foundational models like CLIP, the discrete tokens utilized in VQGAN models show weaker semantic alignment with text, which detracts from

| Method | refCOCOg | | refCOCO+ | | | Reaseg | |
|---|---|---|---|---|---|---|---|
| | val(U) | test(U) | val | testA | testB | gIoU | cIoU |
| ReLA | 65.0 | 66.0 | 66.0 | 71.0 | 57.7 | - | - |
| SEEM | 65.7 | - | - | - | - | 24.3 | 18.7 |
| PixelLM | 69.3 | 70.5 | 66.3 | 71.7 | 58.3 | - | - |
| NExT-Chat | 67.0 | 67.0 | 65.1 | 71.9 | 56.7 | - | - |
| LISA | 67.9 | 70.6 | 65.1 | 70.8 | 58.1 | 47.3 | 48.4 |
| SETOKIM | 71.3 | 71.3 | 68.0 | 72.4 | 61.2 | 50.7 | 52.7 |

Table 4: Results on 3 referring expression segmentation benchmarks. We report cIoU for RefCOCO+/g.

| Mechanism | #Tokens | TFLOPs | Flickr30K | OK-VQA |
|---|---|---|---|---|
| Hard-clustering | 25* | 8.3 | 86.9 | 60.2 |
| Soft-clustering | 23* | 8.2 | 86.7 | 58.9 |
| Fixed | 256 | 15.7 | 85.1 | 51.7 |
| | 64 | 13.9 | 84.1 | 53.6 |
| | 32 | 10.1 | 83.4 | 51.1 |
| | 8 | 8.0 | 82.1 | 50.1 |

Table 5: The effect of different clustering strategies. The first three rows consist of dynamic strategies. #Tokens is the number of tokens, and * denotes the average token number.

| Method | ImageNet | Flickr30K | $VQA^{v2}$ | GQA | MSCOCO |
|---|---|---|---|---|---|
| | (rFID↓) | (CIDEr↑) | (Accuracy↑) | (Accuracy↑) | (FID↓) |
| **SeTok** | 2.07 | 86.9 | 78.5 | 65.6 | 8.5 |
| w/o $\mathcal{L}_{citc}$ | 4.15 | 78.1 | 65.8 | 49.7 | 9.6 |
| w/o PE | 3.56 | 86.1 | 76.2 | 61.4 | 12.8 |
| w/o inter-cluster Transformer | 7.91 | 82.7 | 71.4 | 54.2 | 13.9 |
| w/o inner-cluster Transformer | 6.25 | 85.4 | 73.7 | 53.4 | 11.0 |
| w/o Token Merger | 8.64 | 80.3 | 66.1 | 50.5 | 14.7 |

Table 6: Ablation Study on **SeTok** to image reconstruction, visual understanding, and generation.

their performance in various understanding tasks. Besides, learnable continuous queries transformed via Q-former or cross-attention framework are introduced to alleviate the efficiency issues. However, these methods still struggle with fine-grained semantic alignment with text, potentially limiting the depth of interaction between textual and visual content. By incorporating semantic-equivalent tokens via SeTok, our model secures competitive performances in various vision-understanding tasks. Moreover, our model demonstrates performance improvement on GQA by 3.6%, highlighting our method's superior capability in complex relationships and object quantities reasoning.

**Visual Generation and Editing.** Table 2 demonstrates a comparative analysis of SETOKIM and other diffusion-based and LLM-based methods in vision generation and editing. Notably, compared to other MLLMs integrated with advanced vision decoders such as SD v2.1 (Rombach et al., 2022) and SD-XL (Podell et al., 2024), our method achieves comparable performance on complex prompts. This highlights the effectiveness and efficiency of SeTok in learning the correlations between visual and textual modalities within our unified framework. Further evaluations on instruction-based image editing are conducted. Standard pixel difference (L1), LPIPS (Zhang et al., 2018), and visual feature similarity (CLIP$_{im}$) are employed as metrics. Our model exhibits marked superiority in L1 and CLIP scores compared to existing MLLMs. This enhanced performance can be attributed to SeTok's ability to capture semantically equivalent visual tokens, thereby enhancing the semantic interaction between text and images. Moreover, editing tasks typically involve conceptual replacements within images, and the concept-level token representations learned by our model are inherently well-suited to such tasks involving straightforward replacements or modifications.

**Referring Expression Segmentation.** Table 4 presents MLLMs' performances on referring expression segmentation tasks. Our model consistently outperforms the current SoTA on the RefCOCO+/g and ReaSeg dataset, demonstrating the proficiency of our vision tokens derived from **SeTok** in capturing not only object-centric semantic details but also the high-frequency boundary information.

## 4.2 IN-DEPTH ANALYSIS AND QUALITATIVE EVALUATION

**Ablation Study.** Table 6 summarizes the results of an ablation study evaluating the design benefits of SeTok and the influence of SETOKIM across various vision-language tasks. Firstly, we observe that while the model can achieve commendable reconstruction quality without using contrastive loss, its performance markedly decreases in downstream vision understanding tasks. This suggests that exclusive reliance on reconstruction learning may cause the model to prioritize low-level information at the expense of high-level semantic insights. Furthermore, replacing the token merger with a simple average visual representation for each cluster also results in a significant decline in fine-grained visual understanding and generation performance, possibly due to the averaging process potentially leading to information loss. Lastly, the removal of positional encoding (PE) and both the inner-cluster and inter-cluster transformers degrade the model's performance across various tasks to some extent.

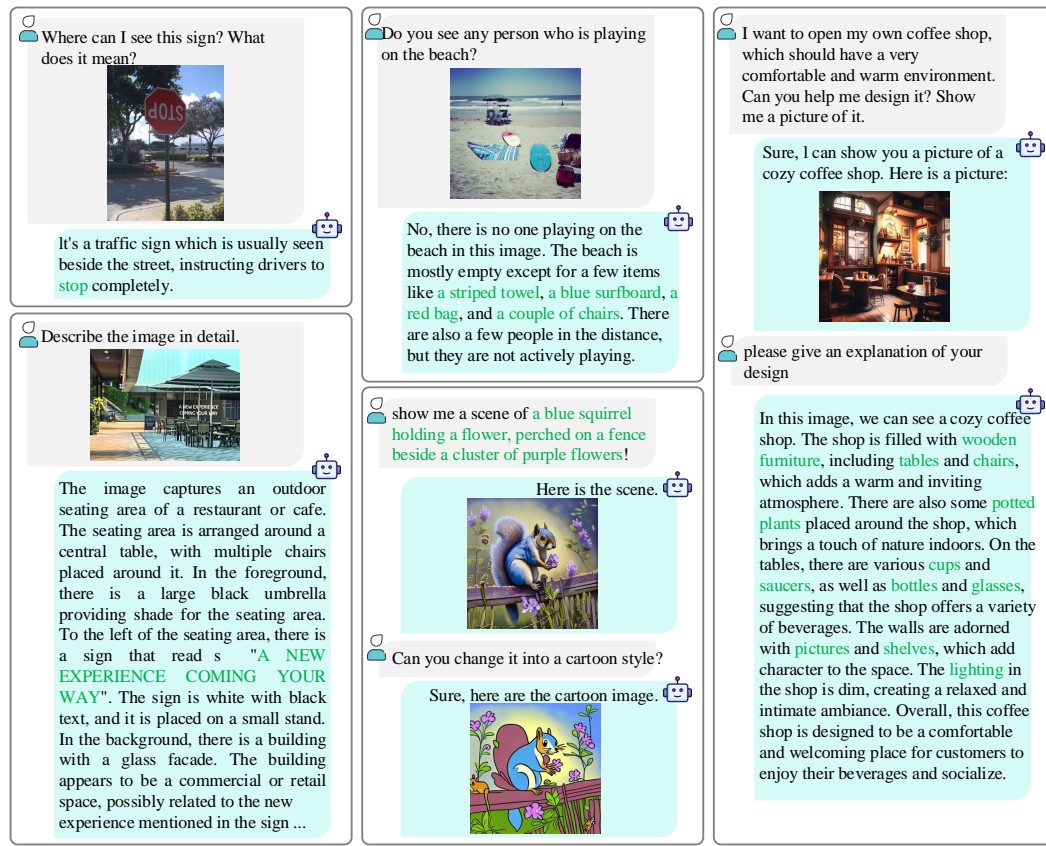

Figure 4: Qualitative results on image understanding and generation. The words marked in green are key elements in questions and answers. Best view it on screen.

**The Impact of the Clustering Mechanism.** Here, we compare the impact of different clustering mechanisms on model performance. As shown in Table 5, we can observe that tokenizers constructed using dynamic clustering mechanisms achieve superior overall performance compared to those with a fixed setup while simultaneously accelerating training time and reducing computational costs during inference. In contrast to soft-clustering, which yields soft attention masks, our findings suggest that hard-clustering produces better results, as it may be because hard clustering leads to higher consistency of cluster outcomes (Haurum et al., 2023), leading to more stable visual tokens and enhancing both the stability and performance of the model. When employing a fixed number of clusters, the critical challenge is to determine the optimal number of clusters. As demonstrated in Table 5, different datasets achieve optimal performance at varying numbers of clusters, with a uniform count across all datasets, resulting in suboptimal outcomes.

**Qualitative Analysis of Visual Understanding and Generation.** As illustrated in Figure 4, our model exhibits proficiency in intricate image understanding tasks, such as deciphering reversed text, exemplified by the word "stop", and accurately identifying text "A NEW EXPERIENCE COMING YOUR WAY" that is partially covered. In tasks involving detailed image descriptions, our approach prioritizes object-level information within images, which substantially mitigates the incidence of hallucinatory responses commonly observed in MLLMs. Moreover, in text-to-image generation, our model demonstrates remarkable capabilities in synthesizing coherent images, which maintain high fidelity and relevance to the textual context, such as the "flower", "fence" and "squirrel".

**Qualitative Analysis of Visual Editing.** Here, we evaluate the efficacy of image manipulation using our model compared to the previous diffusion-based method MagicBrush (Zhang et al., 2024c), and various MLLMs including Emu-2-Gen (Sun et al., 2024a), MGIE (Fu et al., 2024), and Mini-Gemini (Li et al., 2024d). As depicted in Figure 5, SETOKIM displays superior performance by closely adhering to the provided instructions and preserving intricate image details. For instance, our model seamlessly adds "tomato slices" to an image without altering other elements on the pizza,

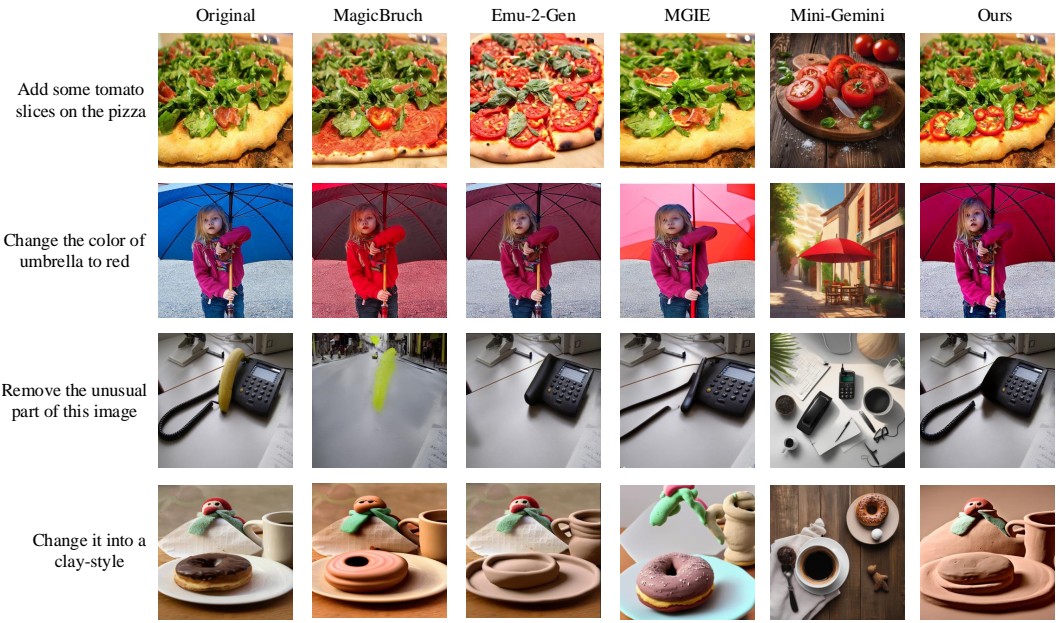

Figure 5: Qualitative comparison between MLLMs for the image editing. SETOKIM excels in adhering to instructions and preserving low-level image details.

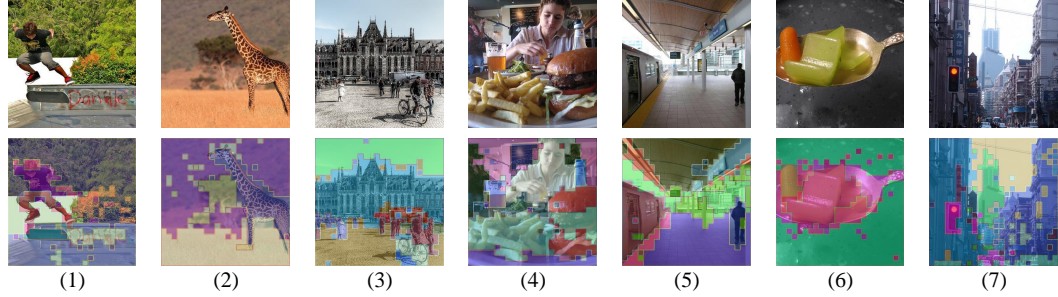

Figure 6: Token mask $M$ visualization of visual tokens generated by **SeTok**.

while Emu-2-Gen and MGIE fall short. Furthermore, our model exhibits remarkable precision in changing the color of an umbrella, while visual objects not intended for alteration retain a high level of consistency before and after editing. Additionally, SETOKIM demonstrates to precisely follow implicit user instructions to remove unusual elements from an image, i.e., the banana, preserving the surrounding context, whereas Emu-2-Gen mistakenly removes a telephone cord and MGIE fails to remove the banana properly, altering the cord's texture. These examples underscore the effectiveness of SETOKIM for high-precision image manipulation, leveraging semantically equivalent visual tokens to achieve nuanced and context-aware results.

**Qualitative Analysis of Visual Tokens.** In Figure 6, we demonstrate how input visual features are assigned to visual tokens after tokenization. First, we observe that our tokenization process resembles partial segmentations, producing semantically complete units. For example, in the second image, visual tokens correspond to distinct elements such as the giraffe, grass, tree, and background, aligning with semantic intuition. Second, the number of tokens obtained from **Setok** is dynamic and not fixed. Third, **SeTok** is capable of adapting to different levels of semantic granularity for the same concept, as seen in images (4) and (5), where the person is represented as a single token. In contrast, in the image (1), the person is divided into tokens for the head, body, and legs. Lastly, in complex scenes, such as the image (7), **SeTok** can still tokenize elements like traffic lights and billboards into semantically complete tokens. Overall, our approach ensures that similar visual features are consistently recognized and processed, improving both coherence and efficiency in tokenization.

## 5 RELATED WORK

Currently, benefiting from the emergent phenomenon, LLMs have demonstrated near-human-level intelligence in language processing (Chiang et al., 2023; Touvron et al., 2023a; Taori et al., 2023). Simultaneously, researchers have been attempting to develop MLLMs by integrating multimodal encoders and decoders into LLMs (Dong et al., 2024; Koh et al., 2023; Lu et al., 2024; Li et al., 2024d; Sun et al., 2024a; 2023; Fei et al., 2024). From the initial MLLMs that could only understand multimodal input signals (Liu et al., 2024b; 2023c) to later versions supporting the generation of multimodal contents (Sun et al., 2023; 2024a; Koh et al., 2023; Wu et al., 2024c), MLLMs have shown powerful capabilities and a broader range of applications. Among all modalities, the integration of vision, known as visual MLLM, has received the most extensive research and application (Gao et al., 2023; Schwenk et al., 2022; Liu et al., 2023b; Lu et al., 2021). The latest MLLM research has not only achieved both understanding and generation of visual content, but also developed more refined, pixel-level visual modeling, including segmentation and editing functions (Yuan et al., 2024; Rasheed et al., 2024; Zhang et al., 2024b; You et al., 2023; Lai et al., 2024).

On the other hand, an increasing body of research indicates that visual tokenization (Dosovitskiy et al., 2021; Ge et al., 2023; Jin et al., 2024b) significantly impacts MLLM capabilities in vision tasks. The fundamental approach involves encoding the input visual content into feature representations via a visual encoder (e.g., Clip-VIT Radford et al. (2021)) and mapping these to an LLM, thus enabling a language-based LLM to understand vision. The corresponding method involves patchifying the original visual images of various sizes into smaller fixed-size patches (Dosovitskiy et al., 2021; Bavishi et al., 2023; Liu et al., 2023c; Sun et al., 2023), treating these as tokens, and encoding each patch/token to obtain corresponding embeddings, which are then fed into the LLM. Subsequent research (Jin et al., 2024b; Ge et al., 2023), aiming further to unify the training objectives of language and visual modalities by introducing codebook techniques, where visual elements are represented as discrete tokens. This allows visual training to be treated similarly to language training, i.e., conducting *next token prediction* (Ge et al., 2023). Unfortunately, whether in the above visual encoding or tokenization techniques, there is a significant bottleneck of MLLM performance: the integrity of visual semantic units, either visual objects or compositional regions, is compromised during the patchifying process. This results in a less effective semantic alignment between vision and language within the LLM. This paper is the first to propose a solution to this problem, introducing a novel Semantic Equivalent Tokenization for MLLM.

In addition, this work is also related to scene decomposition (Yang et al., 2022; Niu et al., 2024; Locatello et al., 2020; Li et al., 2020; 2024b), which involves segmenting a scene into objects. Typically, these methods use a fixed number of query tokens (Kirillov et al., 2023; Suzuki, 2022) and apply cross-attention (Yang et al., 2022; Qi et al., 2023; Li et al., 2024c) to aggregate visual features implicitly. However, this fixed-token approach may not only correspond to the actual visual content but also requires complex network architectures (Caron et al., 2018; Gansbeke et al., 2021) and extensive data for optimization. When combined with LLMs, such complexity significantly increases computational resource demands. Conversely, we learn a dynamic number of semantic objects and do not require complex model structures for optimization, thereby enhancing resource efficiency and providing a more adaptable solution for integrating visual and language modalities.

## 6 CONCLUSION

In this paper, we introduce **SeTok**, a viable semantic-equivalent tokenizer, that enables to tokenize automatically patch-level visual features into a variable number of semantic-complete concept visual tokens. Then, we integrate SeTok with a pre-trained LLM to build an MLLM, SETOKIM, optimized using a unified autoregressive objective and a two-stage training strategy. Extensive experiments demonstrate that our model performs better on a broad range of comprehension, generation, segmentation, and editing tasks, highlighting the effectiveness of **Setok**.

ACKNOWLEDGMENTS

This work is partially supported by NUS Start-up Grant A-0010106-00-00.

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

## A    ETHIC STATEMENT

This work aims to build semantic equivalence tokenization to segment input images into semantic complete tokens to enhance the MLLMs in vision understanding, generation, segmentation, and editing capabilities. Here we discuss all the possible potential impacts of SETOKIM.

**Use of Generative Content**   The SETOKIM, limited by the quantity of fine-tuning data and the quality of the base models, may generate some low-quality content. Also, as a generative model, the LLM will produce hallucinated content in multimodal formats that may be harmful to society. We have reminded users to interpret the results with caution. Anyone who uses this LLM should obey the rules in a license. And also commercial use of our system is not allowed.

**Data Privacy and Security**   Our research utilizes datasets that are either publicly available or collected with explicit consent. We adhere to strict data privacy and security protocols to protect the information and ensure it is used solely for this research.

**Bias Mitigation**   Recognizing the potential for bias in AI models, particularly in vision-language tasks, we rigorously test our tokenizer across diverse datasets. This approach is designed to identify and mitigate biases that may affect the model's performance or lead to unfair outcomes in its applications.

## B    LIMITATION

While SETOKIM has achieved further improvements across various language-driven vision tasks, becoming a zero-shot general specialist, it still faces several limitations.

**Model Scale.**   The evaluation of our model is currently constrained to configurations with 7B parameters. As shown in (Laurençon et al., 2024), the performance of MLLMs is limited by the scale of the core backbone LLM. Despite the impressive results achieved, the potential benefits of employing significantly larger models, such as 65B or 130B, are worth exploring in future studies.

**The Resolution of Image.**   Our model supports images with resolutions up to $384 \times 384$, enabling the understanding of visually fine-grained content. While there have been improvements in understanding visually fine-grained content, challenges remain when processing higher-resolution images, particularly for tasks requiring detailed visual reasoning. Recent advancements have explored various strategies to address these challenges. For instance, Shi et al. (2024) highlights that straightforward channel concatenation between low- and high-resolution features serves as an efficient and effective fusion strategy, achieving a balance between performance and computational efficiency. Moreover, the use of mixture-of-experts (MoE) structures has shown significant improvements when combining different vision encoders. Despite these advances, there is still a need to enhance the understanding of low-resolution inputs and the ability to generalize across diverse modalities, particularly for tasks where fine-grained details are embedded in low-resolution visual data.

**Hallucination.**   Although our model has made some progress in mitigating hallucination through fine-grained vision-language alignment, as demonstrated in experiments on the POPE dataset, hallucinations remain inevitable. This area continues to pose challenges and is crucial for future exploration and enhancement.

## C    DETAILED METHOD

### C.1    TOKEN CLUSTER

The formal token clustering algorithm is described in Algorithm 1. Specifically, a scope $z = [0, 1]^{h \times w}$ is initialized to a matrix of ones $\mathbf{1}^{h \times w}$ to track the degree to which visual embeddings have been assigned to clusters. In addition, the seed scores are initialized by combining the local density in Eq.(1) and distance in Eq.(2) to perform the selection of visual embeddings. At each iteration, a

single embedding vector $\boldsymbol{x}_{i,j}$ is selected at the spatial location $(i, j)$ which corresponds to the argmax of the element-wise multiplication of the seed scores and the current scope. This ensures that cluster seeds are sampled from pixel embeddings that have not yet been assigned to clusters. An alpha mask $\alpha_c \in [0, 1]^{h \times w}$ is computed as the distance between the cluster seed embedding $\boldsymbol{x}_{i,j}$ and all individual pixel embeddings according to a distance kernel $\varphi$. The output of the kernel $\varphi$ is one if two embeddings are identical and decreases to zero as the distance between a pair of embeddings increases. Additionally, a negative penalty $\beta\boldsymbol{s}$ is applied to the alpha mask by misusing the seed scores, where $\beta$ is a hyper-parameter. This encourages the selection of elements similar to the current feature with lower information density. The associated concept mask $\boldsymbol{M}_c$ is obtained by the element-wise multiplication of the alpha masks by the current scope. An element-wise multiplication with the complement of the alpha masks then updates the scope. This process is repeated until a stopping condition is satisfied, at which point the final scope is added as an additional mask to explain any remaining embeddings.

---

**Algorithm 1** Token Clustering Algorithm

---

**Require:** visual embeddings $\boldsymbol{X} \in \mathbb{R}^{h \times w \times d}$
**Ensure:** masks $\boldsymbol{M} \in [0, 1]^{h \times w \times C}$ with $\sum_c M_{i,j,c} = 1$
 1: **Initialize**: masks $\boldsymbol{M} = \emptyset$, scope $\boldsymbol{z} = \boldsymbol{1}^{h \times w}$, seed scores $\boldsymbol{s} \in \mathbb{R}^{h \times w}$
 2: **while** not StopCondition($\boldsymbol{M}$) **do**
 3:     $(i, j) = \arg\max(\boldsymbol{z} \odot \boldsymbol{s})$
 4:     $\alpha = \text{sigmoid}(\varphi(\boldsymbol{X}, (i, j)) - \beta\boldsymbol{s})$
 5:     $\boldsymbol{M}.\text{append}(\boldsymbol{z} \odot \alpha)$
 6:     $\boldsymbol{z} = \boldsymbol{z} \odot (1 - \alpha)$
 7: **end while**
 8: $\boldsymbol{M}.\text{append}(\boldsymbol{z})$

---

### C.2 Concept-level Image-text Contrastive Loss

To enable effective visual concept token learning, we propose concept-level image-text contrastive loss. Specifically, we randomly select K objects in the image, and acquire the corresponding object labels, and then prompt each of them with a set of handcrafted sentence templates, e.g., 'A photo of a {object label}'. The motivation for selecting objects is that they are the smallest units of image representation with complete semantics and have a corresponding relationship with the semantic units in the text. Next, we employ contrastive losses between the new sets of image-'prompted text' pairs $\{(I, T_1), (I, T_2), \cdots, (I, T_K)\}$ where $\{T_k\}_{k=1}^K$ are all prompted sentences generated from the objects sampled from the image $I$. Among the batch $B$, each image has $K$ positive text pairs and $B(K - 1)$ negative pairs. Similarly to the standard image-text contrastive loss (Radford et al., 2021), we define the concept-level image-text contrastive loss as a sum of two two-way contrastive losses:

$$\mathcal{L}_{I \to \{T_k\}_{k=1}^K} = -\frac{1}{B} \sum_{i=1}^B \log \frac{\sum_{k=1}^K \exp(\boldsymbol{V}_i^I \cdot \boldsymbol{V}_i^{T_k}/\tau)}{\sum_{k=1}^K \sum_{j=1}^B \exp(\boldsymbol{V}_i^I \cdot \boldsymbol{V}_j^{T_k}/\tau)}, \tag{5}$$

$$\mathcal{L}_{\{T_k\}_{k=1}^K \to I} = -\frac{1}{B} \sum_{i=1}^B \log \frac{\sum_{k=1}^K \exp(\boldsymbol{V}_i^{T_k} \cdot \boldsymbol{V}_i^I/\tau)}{\sum_{k=1}^K \sum_{j=1}^B \exp(\boldsymbol{V}_j^{T_k} \cdot \boldsymbol{V}_i^I/\tau)}, \tag{6}$$

$$\mathcal{L}_{I \leftrightarrow \{T_k\}_{k=1}^K} = \mathcal{L}_{I \to \{T_k\}_{k=1}^K} + \mathcal{L}_{\{T_k\}_{k=1}^K \to I}, \tag{7}$$

where the concept representation $\boldsymbol{V}_i^{T_k}$ is extracted by the pre-trained CLIP-based text encoder, which is frozen during training.

## D Detailed Experiments Settings

### D.1 Implementation Details

For the **SeTok**, we apply pre-trained SigLIP-SO400M-patch14-384 (Zhai et al., 2023) as our vision encoder, and the numbers of inner-cluster and inter-cluster transformer layers are set as 12, and 8,

| | Name | Size |
|---|---|---|
| **SeTok** | ImageNet-1K (Deng et al., 2009) | 1.2M |
| | OpenImages (Kuznetsova et al., 2020) | 9M |
| **Stage-I** | CC12M (Changpinyo et al., 2021) | 12M |
| | LAION-aesthetics-12M (Schuhmann et al., 2022) | 12M |
| | ALLaVA-Caption-4V (Chen et al., 2024) | 715K |
| | InstructPix2Pix (Brooks et al., 2023) | 313K |
| | LLaVA-595K (Liu et al., 2023c) | 595K |
| | MSCOCO (Lin et al., 2014) | 313K |
| | Visual Genome (Krishna et al., 2017) | 108K |
| | OpenImages (Kuznetsova et al., 2020) | 9M |
| | SlimPajama (Soboleva et al., 2023) | - |
| **Stage-II** | ALLaVA-Instruct-4V (Chen et al., 2024) | 661K |
| | ShareGPT4V (Krishna et al., 2017) | 80K |
| | Alpaca (Taori et al., 2023) | 5K |
| | LLaVA-v1.5-mix-665K (Liu et al., 2023c) | 665K |
| | VQA$^{v2}$ (Goyal et al., 2019) | 83K |
| | GQA (Hudson & Manning, 2019) | 72K |
| | OKVQA (Marino et al., 2019) | 9K |
| | AOKVQA (Schwenk et al., 2022) | 50K |
| | RefCOCO/+/g (Kazemzadeh et al., 2014; Mao et al., 2016) | 65K |
| | InstructPix2Pix (Brooks et al., 2023) | 313K |
| | MagicBrush (Zhang et al., 2024c) | 10K |

Table 7: The training data used in our experiments.

respectively. The dimension of the semantic-equivalent token is 512. For the detokenizer, we adopt $L = 12$ transformer-based layers with cross-attention, where the keys and values are derived from a fixed number of masked tokens. This process converts the dynamic number of tokens into a fixed-size representation. Also, inspired by Yu et al. (2024), we employ a CNN-based pixel decoder with an upsampler to reconstruct the original images.

In the **SETOKIM** framework, we employ the LLaMA-2-7B (Touvron et al., 2023b) to initialize our LLM backbone. Following Kirillov et al. (2023), we take the image embedding extracted in the vision encoder in **SeTok** and the visual tokens generated by LLM as inputs, which are both fed into the mask decoder. This decoder uses prompt self-attention and cross-attention in two directions (prompt-to-image embedding and vice-versa) to update all embeddings. After running two blocks, we upsample the image embedding and an MLP maps the output token to a dynamic linear classifier, which then computes the mask foreground probability at each image location. Following Li et al. (2024a), we employ a small MLP consisting of three residual blocks (He et al., 2016) for computing the diffusion loss. Each block sequentially applies a LayerNorm (LN) (Ba et al., 2016), a linear layer, SiLU (Elfwing et al., 2018), and another linear layer, merging with a residual connection.

## D.2 TRAINING DATA

Here, we detail the training data utilized for training **SeTok** and **SETOKIM** in Table 7. In the training phase of **SeTok**, ImageNet-1K (Deng et al., 2009) is employed for reconstruction tasks, while OpenImages (Kuznetsova et al., 2020) supports both reconstruction and alignment learning. Additionally, some overlap exists between datasets used in **Stage-I** and **Stage-II** training. For instance, datasets like VQA$^{v2}$ (Goyal et al., 2019), ShareGPT4V (Krishna et al., 2017), and GQA (Hudson & Manning, 2019) have been included in LLaVA-v1.5-mix-665 (Liu et al., 2023c). To provide a clear and comprehensive view of the training data sources and their usage, we explicitly enumerate all datasets included in the training pipeline.

## D.3 TRAINING RECEIPT

In Table 9, we list the detailed hyper-parameters setting at three stages, i.e., **Setok** training and two-stage **SETOKIM** training. All training is conducted on 64× H100 (80G) GPUs.

| Model | LLM | Vision Encoder | Image Resolution | Data Size | |
|---|---|---|---|---|---|
| | | | | Pretrain | Finetune |
| InstructBLIP | Vicuna-13B | ViT-g/14 | 224 | 129M | 1.2M |
| Qwen-VL-Chat | Qwen-7B | ViT-bigG (Fine-tuned) | 448 | 1.4B | 50M |
| Emu | LLaMA-7B | EVA-01-CLIP | 224 | >600M | 312K |
| DreamLLM | Vicuna-7B | CLIP L/14 | 224 | 32M | 120K |
| LLaVA-1.5 | Vicuna-1.5 7B | CLIP ViT-L/336px | 336 | 558K | 665K |
| SEED-X | Llama2-chat-13B | Qwen-VL | 448 | 158M | >50M |
| LaVIT | LLaMA-7B | ViT-G/14 of EVA-CLIP | 224 | 100M | 193M |
| Unified-IO-2 | - | ViT-B | 384 | 1.127B | 559M |
| CM3Leon | - | VQVAE | 256 | 2.4T tokens | 11.4M |
| Chameleon | - | VQVAE | 512 | >1.4B | 1.8M |
| SETOKIM | Llama2-7B | SigLIP-SO400M-patch14-384 | 384 | 35M | 1.2M |

Table 8: Configuration comparison between baselines and SETOKIM. "-" indicates training the LLM from scratch.

## D.4 BASELINES.

Here, we explicitly demonstrate a configuration comparison in terms of the LLM version, vision encoder, and data size used in the baselines and SETOKIM in Table 8.

| Configuration | SeTok | Stage-I | Stage-II |
|---|---|---|---|
| Optimizer | AdamW | AdamW | AdamW |
| Precision | bfloat16 | bfloat16 | bfloat16 |
| Peak learning rate of LLM | - | 5e-5 | 5e-5 |
| Peak learning rate of Visual Part | 5e-4 | 1e-4 | 2e-4 |
| Weight Decay | 0.05 | 0.1 | 0.01 |
| Learning Rate Scheduler | Cosine | Cosine | Cosine |
| LR Warmup Steps | 10K | 2K | 5K |
| Input image resolution | 384×384 | 384×384 | 384×384 |
| Batch Size Per GPU | 16 | 16 | 16 |
| Gradient Accumulation Steps | 8 | 8 | 8 |
| Maximum Token Length | - | 2048 | 2048 |

Table 9: Training recipes for **SeTok**, SETOKIM of Stage-I: Multimodal Pretraining and Stage-II: End-to-end Instruction Tuning.

## E EXTENDED EXPERIMENTAL ANALYSIS

| Setting | lr-v | lr-t | Text | Multi-modal | Humanities | STEM | Social Sciences | Other | Average |
|---|---|---|---|---|---|---|---|---|---|
| LLaMA-2-7B | - | 3e-4 | 100% | 0% | 42.9 | 36.4 | 51.2 | 52.2 | 45.3 |
| SETOKIM | 1e-4 | 5e-5 | 70% | 30% | 41.7 | 34.8 | 49.4 | 51.0 | 43.9 |
| SETOKIM | 1e-4 | 5e-5 | 50% | 50% | 37.5 | 31.4 | 46.3 | 45.9 | 40.1 |
| SETOKIM | 1e-4 | 5e-5 | 30% | 70% | 30.3 | 31.7 | 44.7 | 41.1 | 35.4 |

Table 10: LLM comparison by varying the language-vision dataset ratio.

**The Impact of Language Volume.** Before performing Stage-2 instruction training, we conduct experiments with mixing text and image data in various proportions to identify the optimal balance of additional text data. The experimental results on the MMLU dataset are summarized in Table 10. Our findings suggest that a ratio of 7:3 (Language:Vision) is optimal, as it minimally impacts the LLM's language performance (-1.4 on MMLU) while achieving the best results on both multimodal understanding and generation tasks.

| Method | Flickr30K (CIDEr↑) | VQAv2 (Accuracy↑) | GQA (Accuracy↑) |
|---|---|---|---|
| SeTok | 86.9 | 78.5 | 65.6 |
| w/ $\mathcal{L}_{rec}$ | 78.1 | 65.8 | 49.7 |
| w/ $\mathcal{L}_{cite}$ | 83.6 | 76.3 | 63.4 |

Table 11: The effect of unlocking vision encoder in training **Setok** and **Setokim**.

**The Loss Impact for Setok.** We argue that a reasonable tokenizer must possess two essential attributes: **1)** Complete and enriched high-level semantic information and **2)** Undistorted pixel-level details. Therefore, we design to optimize the Setok by minimizing the reconstruction loss and concept-level image-text contrastive loss. Here, we conduct further experiments to explore the effect of each loss on tokenizer performance. As the results shown in Table 11, we observe that the performance with only $\mathcal{L}_{cite}$ is superior to that with only $\mathcal{L}_{rec}$. We attribute this to the fact that relying solely on $\mathcal{L}_{rec}$ causes the tokenizer to focus primarily on pixel-level information, often at the neglect of high-level semantic information. This imbalance may introduce challenges for the LLM when interpreting image semantic content with limited training data.

| Setting | ImageNet (rFID↓) | Flickr30(CIDEr.↑) | $VQA^{v2}$ (Acc.↑) |
|---|---|---|---|
| Frozen | 123.6 | 85.4 | 77.5 |
| UnFrozen | 2.07 | 86.9 | 78.7 |

Table 12: The effect of unlocking vision encoder in training **Setok** and **Setokim**.

**The Impact of Unfreeze Vision Encoder.** To evaluate the impact of unfreezing the vision encoder, we conduct an ablation experiment where the vision encoder is kept frozen, and only the token merger and detokenizer are optimized. We observe that **SeTok** fails to reconstruct the image as freezing the vision encoder hinders its ability to learn the low-level features required for accurate reconstruction. In this scenario, the vision decoder alone is tasked with reconstruction, but it is unable to do so effectively using only high-level semantic features. Interestingly, freezing the vision encoder did not noticeably impact **SeTok**'s performance in vision-language semantic understanding.

| Mechanism | #Tokens | TFLOPs | Flickr30K | $VQA^{v2}$ | OK-VQA |
|---|---|---|---|---|---|
| SigLIP + MLP (Liu et al., 2024b) | 256(Fixed) | 15.8 | 80.6 | 72.4 | 56.1 |
| SigLIP + Q-former (Zhu et al., 2023) | 32(Fixed) | 12.4 | 81.3 | 71.0 | 54.6 |
| SigLIP + Resampler(Alayrac et al., 2022) | 64(Fixed) | 13.4 | 83.4 | 72.5 | 54.9 |
| SeTok | Dynamic | 8.2 | 86.9 | 78.7 | 60.2 |

Table 13: Comparison between **Setok** and other vision tokenization approaches, all of which generate continuous visual tokens that are subsequently fed into the LLM.

**The Comparison of Vision tokenizer.** To evaluate whether our proposed **SeTok** effectively integrates with LLMs to enhance model performance, we experimented with different connector strategies, such as MLP (Liu et al., 2024b), Q-former (Zhu et al., 2023) and Resampler (Alayrac et al., 2022). Using the same vision encoder (i.e., SigLIP-SO400M-patch14-384), we construct various MLLM architectures. We follow a two-stage training process on the same dataset. Finally, we assessed the models' performance on vision-languages tasks, and the results are presented in Table 13. As observed, SeTok demonstrates higher efficiency, achieving lower TFLOPS while delivering superior vision understanding capabilities. These findings validate that SeTok is capable of learning more aligned and compact visual tokens, leading to better semantic integration and improved performance.

Furthermore, we retrained **Setokim** using the same dataset as LLaVA-1.5, focusing solely on performance in visual understanding tasks. As shown in Table 14, our model consistently outperforms LLaVA across benchmarks, highlighting **Setok**'s ability to achieve more effective vision-language alignment and enhance overall performance.

**Qualitative Analysis of Visual Segmentation.** We present the segmentation examples in Figure 8. It is easy to note that the attention mask closely aligns with the object mask, and our model

| Method | VQA$^{v2}$ | GQA | VisWiz | POPE | MME | MM-Vet |
|--------|-----------|------|--------|------|--------|--------|
| LLaVA-1.5 | 78.5* | 62.0* | 50.0 | 85.9 | 1510.7 | 33.1 |
| SETOKIM | 78.6* | 63.8* | 52.7 | 87.6 | 1521.4 | 40.3 |

Table 14: Comparison between SETOKIM and LLaVA using the same dataset for training. *: indicate the training datasets are observed during training.

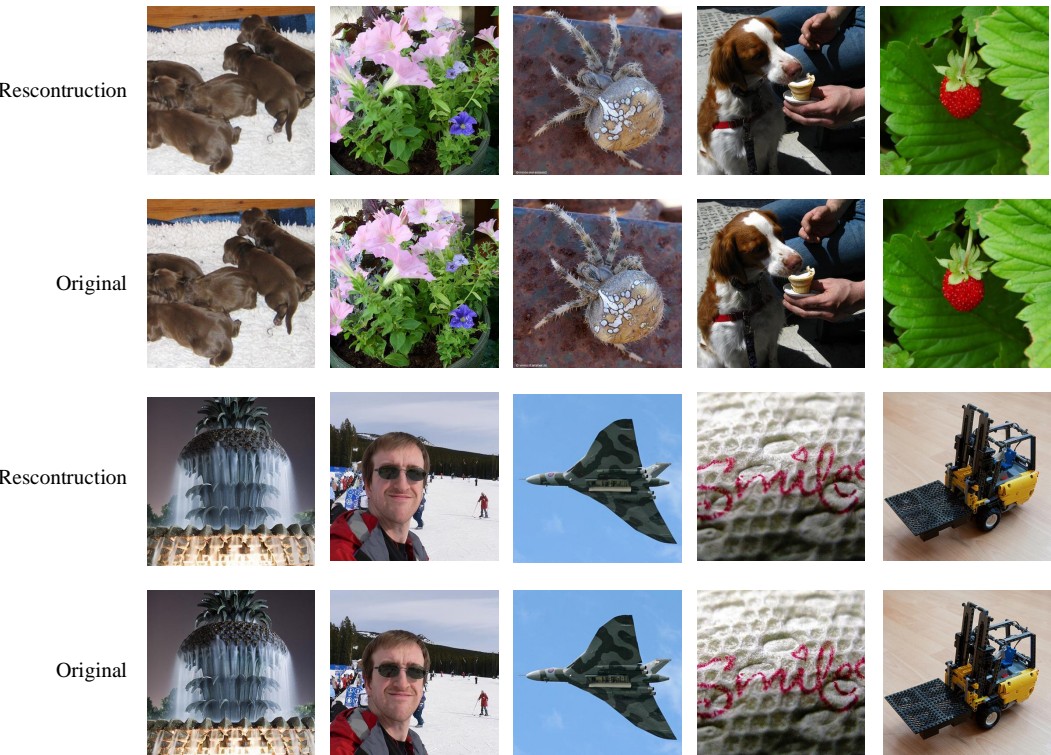

Figure 7: The image reconstruction results from the visual detokenizer in **Setok**.

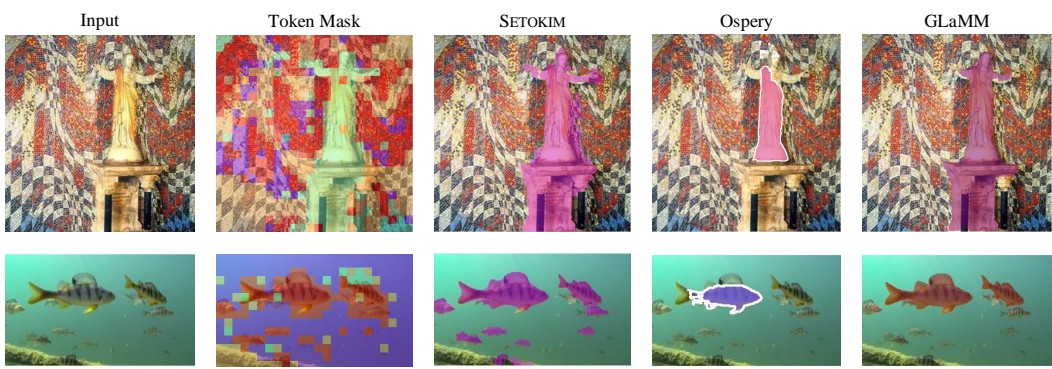

Figure 8: The visualizations for segmentation results compared with GLaMM (Rasheed et al., 2024) and Osprey (Yuan et al., 2024).

shows superiority in achieving more accurate and detailed segmentation results than other LLM-based segmentation methods. Notably, as depicted in the second row of this figure, the visual token generated by our method encompasses all depicted fish, effectively achieving a complete segmentation of the fish in the scene. In contrast, other models produce only partial segmentation. This effectiveness of the segmentation highlights the precise content representation and improved interpretability of the visual tokens. Such visual tokens can eventually enhance the vision-language understanding incorporated with the text tokens.

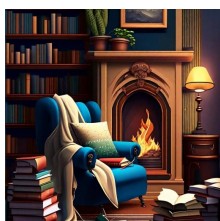
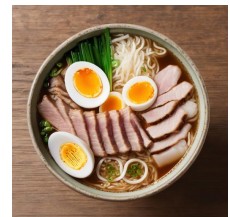
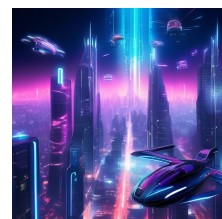
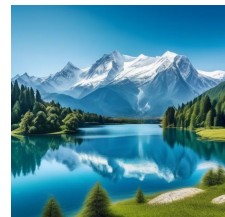

A cozy library filled with books, with an armchair in the corner and a fireplace flickering nearby.

A bowl of ramen with steaming broth, slices of pork, soft-boiled eggs, and green onions, presented in a traditional Japanese setting.

A futuristic city with tall, glowing skyscrapers and flying cars zooming through the sky at night, with neon lights illuminating the streets.

A majestic mountain range with snow-capped peaks under a clear blue sky, surrounded by lush green forests and a crystal-clear lake.

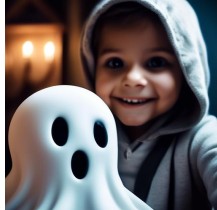
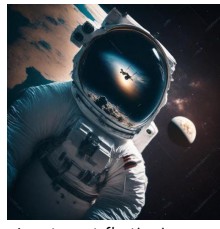
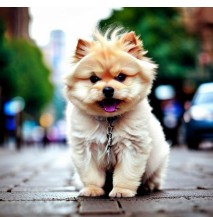
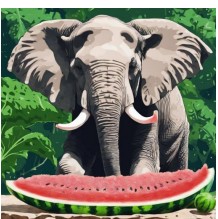

A selfie of a child with a cute ghost.

An astronaut floating in space, gazing at the Earth with stars twinkling in the background.

A dog is sitting on the street.

An elephant is eating watermelon.

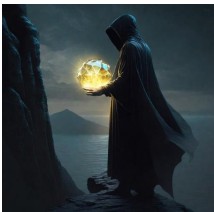
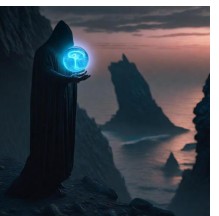
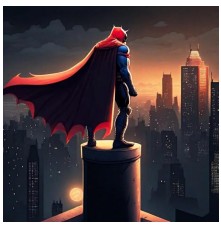

A mysterious figure in a dark cloak, holding a glowing crystal in one hand while standing at the edge of a cliff.

A superhero standing on the rooftop of a tall building, cape billowing in the wind, watching over the city at night.

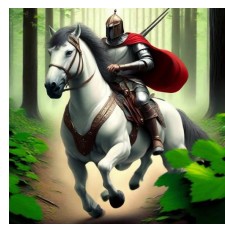
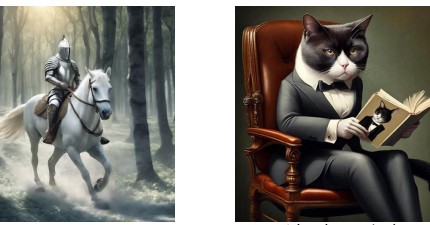
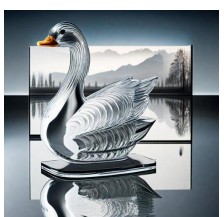

A brave knight riding a white horse through an enchanted forest.

A cat with a human body wear a tuxedo is sitting on a chair, reading a book.

A transparent sculpture of a duck made out of glass. The sculpture is in front of a painting of a landscape.

Figure 9: The visualization of generation images from **SETOKIM**.

**The Quantitative Reconstruction of SeTok.** In Figure 7, we visualize some reconstructed examples by Setok. It can be seen that, given the tokenized visual tokens, the original input images can be successfully recovered. The reconstructed examples exhibit a high degree of the construction of the method.

**Visual Generations.** Figure 9 visualizes the images generated by **SETOKIM**.

**Visual Understanding.** Figure 10 presents additional examples of vision-language understanding and reasoning tasks. Notably, as shown in Figure 11, **SETOKIM** exhibits strong in-context learning and multi-image reasoning capabilities.

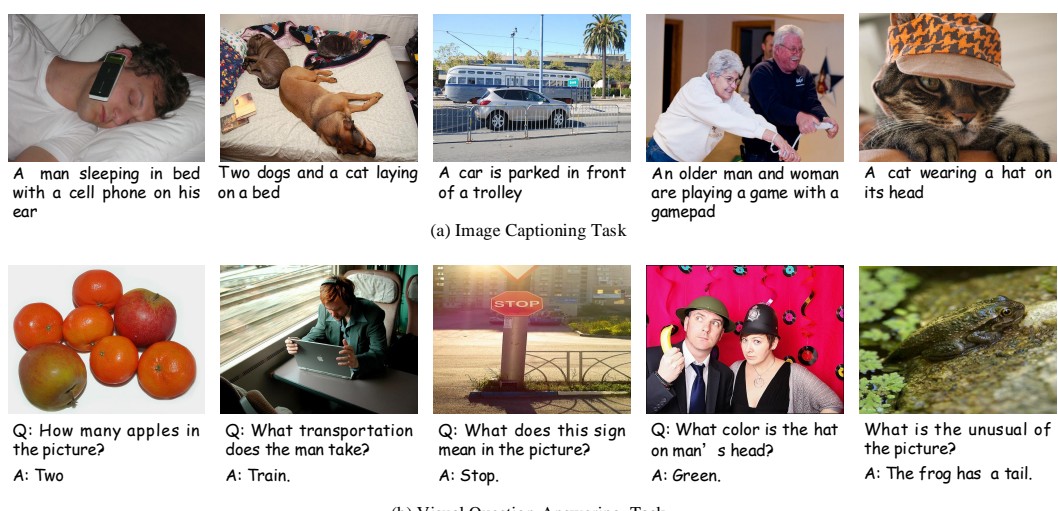

(a) Image Captioning Task

(b) Visual Question-Answering Task

Figure 10: The **SETOKIM**'s performance visualization of image captioning (a) and VQA (b) task.

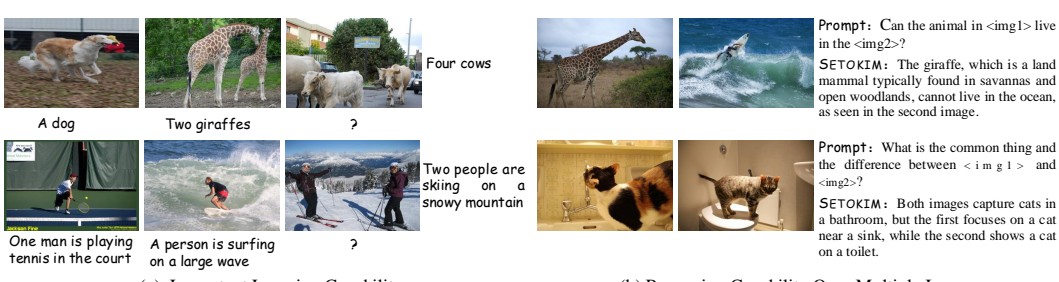

(a) In-context Learning Capability      (b) Reasoning Capability Over Multiple Images

Figure 11: Illustration of **SETOKIM** performing in-context learning in (a) with two image-text pairs and a third image as context to prompt the model, and reasoning across multiple images in (b) with two images with the question as context to guide the model.

