# OpenReview forum: "Towards Semantic Equivalence of Tokenization in Multimodal LLM"
_ICLR.cc/2025/Conference — ICLR 2025 Poster_

### Official Review · Reviewer_zdxE · 2024-10-31

**Soundness:** 2
**Presentation:** 3
**Contribution:** 3
**Rating:** 6
**Confidence:** 5

**Summary:**

This paper introduces one viable semantic-equivalent tokenizer for MLLMs, namely, SeTok. It tokenizes automatically patch-level visual features into a variable number of semantic-complete concept visual tokens, further achieving accurate alignment between vision and text.
And then use a unified autoregressive objective to perform two-stage optimization on SETOKIM (one MLLM). Experiments covering comprehension, generation, segmentation, and editing tasks, demonstrate the effectiveness.

**Strengths:**

[+] For MLLMs, the pursuit of a more reasonable tokenizer to unify visual tasks and generate visual tokens that are accurately aligned with text semantics has always been a key challenge that deserves further research.

[+] The paper is easy to follow and understand, having clear formulation.

[+] Many experiments are conducted to demonstrate the idea’s performance, across comprehension, generation, segmentation, and editing tasks, showing good generality.

**Weaknesses:**

[-] Complex images. SeTok seems to be effective in handling simple images, but when faced with complex images, the quality of clustering seems difficult to guarantee (for example, there are dense people, objects, or significant repetition between people and objects in the image). As a result, clustering errors are injected into visual tokens, which corrupt the final output results.

[-] Source of gains. For MLLMs, there are usually two types of visual bottlenecks: A. maintaining detail resolution to form hierarchical information [1]; B. one reasonable tokenizer. When demonstrating the value of the tokenizer, this paper still uses the detokenizer (in Fig. 2) to reconstruct pixel level images, i.e., keeping visual details. Can you decouple these two to demonstrate the effectiveness of each component separately? For example, report the performance with only \mathcal{L}_{citc}, or make more comparisons with [1].

[1] Eagle: Exploring the design space for multimodal llms with mixture of encoders

**Questions:**

[-] Training/inference efficiency. Due to the clustering of visual tokens, the number of tokens obtained from each image is dynamic. So, how to perform parallel training on a batch, without affecting the efficiency of training and testing due to clustering?

---

### Official Review · Reviewer_JdMJ · 2024-11-03

**Soundness:** 3
**Presentation:** 3
**Contribution:** 3
**Rating:** 6
**Confidence:** 4

**Summary:**

In this paper, the authors propose a dynamic Semantic-Equivalent Vision Tokenizer (SeTok) to preserve semantic integrity in visual features. SeTok can produce different numbers of visual tokens for various images. The authors introduce concept-level image-text contrastive loss and image reconstruction loss to balance the high-level understanding and fine-grained detail preservation abilities of SeTok. SeTokIM, an MLLM equipped with SeTok, shows superior performances on image-text understanding and generation tasks.

**Strengths:**

1. The paper is well-written and easy to follow.
2. The experiments are well-conducted and quite comprehensive.
3. The ablation studies of SeTok is quite clear and valuable, highlighting the effectiveness of the proposed components.
4. I think that the use of dynamic clustering to generate semantic-equivalent visual tokens in MLLM is quite interesting, compared to standard operations like average pooling and query-based compression. The comparison of dynamic and fixed clustering demonstrates the superiority of this clustering mechanism.

**Weaknesses:**

I have some concerns regarding the comparisons in Table 1. The authors claim that SeTok is superior to Q-former or cross-attention framework. However, SeTokim and the other compared MLLMs do not share the same training scheduler, base ViT, LLM, or training datasets. A fair comparison is to evaluate various kinds of visual tokenizers in an environment with tightly controlled variables.

**Questions:**

1. Why do we choose ImageNet-1K and OpenImages for reconstruction and alignment learning separately?
2. How to do fixed clustering? k-means?

---

### Official Review · Reviewer_QXWj · 2024-11-04

**Soundness:** 3
**Presentation:** 2
**Contribution:** 3
**Rating:** 5
**Confidence:** 5

**Summary:**

This paper focuses on the design of visual tokenizer for MLLMs. Current MLLMs do not consider the visual semantic integrity when processing images into patches. To address this, the authors propose SeTok that employs a dynamic clustering strategy to groups visual features into semantic units. Furthermore, they integrate SeTok into MLLM, creating SeTokIM, and demonstrate promising results on both generation and understanding tasks.

**Strengths:**

1. The proposed SeTok is interesting and valuable.
2. The experimental details in the paper are thorough, ensuring reproducibility.

**Weaknesses:**

1. The performance of MLLMs is greatly influenced by SFT data and LLMs. Therefore, when comparing image understanding capabilities, it is important to make sure under the same settings, such as using the same settings as LLaVA, to better reflect the effectiveness of the proposed methods. However, there is lack of this comparison in the experiments.
2. In the comparison of image understanding capabilities, there is a lack of evaluation on more fine-grained benchmarks such as TextVQA, OCRBench, SEED-Bench-2-Plus, as well as more general benchmarks like MMBench and SeedBench. A potential concern is that SeTok's dynamic clustering strategy might lead to the loss of fine-grained information, making it challenging to complete more difficult tasks.

**Questions:**

1. Is SeTok trainable during multimodal pre-training stage?
2. How much additional text data needs to be added to avoid degrading the performance of LLMs?

---

> ### Author Response · Authors · 2024-11-26
> **Please let us know whether all issues are addressed**
>
> Dear reviewer,
>
> Thanks for the comments and review. We have provided more explanations and answers to your questions. Since the deadline for discussion is near the end, please let us know whether we have answered all the questions. If you have more questions, please raise them and we will reply ASAP.
>
> Thanks, Authors

---

### Official Review · Reviewer_sWMT · 2024-11-05

**Soundness:** 2
**Presentation:** 4
**Contribution:** 3
**Rating:** 8
**Confidence:** 4

**Summary:**

This paper proposes a new visual tokenization method SeTok, which groups visual features into semantic units via a dynamic clustering algorithm. This design is aimed to address the information fragmenting problem and the semantic alignment problem in previous visual tokenization methods. A MLLM called SETOKIM is trained based on the proposed tokenization method.

**Strengths:**

- The motivation of dynamic SeTok makes a lot of sense. As the paper mentioned, fragment visual input can corrupt the visual semantic integrity, which is a long-lasting problem in visual tokenization. This paper provides a simple solution though clustering but exhibit high performance, which could inspire future works.
- Sufficient experimental evaluation. Various tasks are included, like referring expression segmentation, visual understanding, and text-to-image generation. This could take much efforts. Also, this paper present visualization of the token masking results of SeTok in Fig. 7, which I think is a strong support for the motivation.

**Weaknesses:**

- Unfair comparison. 1) The model is finetuned on a lot of multimodal instruction datasets, whereas the compared works, like the ones in Tab. 1, use smaller data sizes for finetuning. Performing comparison on different training data hardly demonstrates the priority. 2) The visual encoder introduces more parameters for clustering (totally 20 transformer layers). 3) A more advancing pretrained encoder SigLIP-384 is used. In conclusion, many settings, including the data, the model parameters, and the pretrained encoder, exist unfair comparison. These settings conceal the true effectiveness of the proposed tokenization method. I understand that it is hard for an MLLM to perfectly align with previous settings, and it is ok. But my intension is to make sure the authors can explicitly present these differences in the tables, which helps prevent misleading and ensure fair comparisons in future research.

- Potential data leak. The benchmarks used in Tab. 1 all have released answers, which may be potentially seen during training. It is suggested to perform additional evaluation on some addition benchmarks that do not have released answers, like MMB.

- Potentially poor performance on fine-grained visual perception. The clustering-based visual tokenization may be unfriendly to some fine-grained, like OCR, because clustering always fuses information for coarse representation.

**Questions:**

- How is the visual detokenizer trained? As the tokenized visual tokens have dynamic numbers, it is very confused for me on how the visual tokenizer is trained based on the dynamic visual tokens. And I found no description on this in the paper.

- I also have an open question for the author. Any answers should be ok. The proposed tokenization method groups the vision inputs without any semantic instruction. But as we know, vision has infinite divisibility, which means that all objects can always be decomposed into smaller parts. Based on this, maybe a more reasonable way is to perform clustering during the reasoning process inner the MLLM according to corresponding contexts, instead of the input part. So, I would like to hear the authors’ thoughts on this.

I intend to raise my score if the authors can properly address my concerns in the weakness.

---

### Official Review · Reviewer_zAwn · 2024-11-09

**Soundness:** 3
**Presentation:** 3
**Contribution:** 3
**Rating:** 6
**Confidence:** 4

**Summary:**

This work purpose a novel dynamic Semantic-Equivalent Vision Tokenizer(SeTok)，using adynamic cluster method to group visual features into semantic units, , flexibly determining the number of tokens based on image complexity. The proposed SETOK-based MLLM (SETOKIM) shows superior performance on various text-image tasks.

**Strengths:**

1.This paper points out that the main bottleneck of current MLLMs is visual tokenization method, which cannot effectively align semantic and image features. Current visual tokenization methods only divide the image into a fixed number of patches, which not only destroys the visual context information, but also makes it difficult to align the visual semantic units.

2.During SeTok training, this work introduces concept-level text-image contrastive loss and image reconstruction loss. The former ensures the semantic independence and integrity of each patch, and semantically alignes visual tokens with the corresponding text concepts. The latter ensures that the token retains enough pixel-level detail by calculating the loss when feeding the token into the detokenizer to reconstruct the original image. Finally, the sum of the two losses is calculated using weighting.

3.The ablation experiments of key modules are sufficient, and the performance of the method on various language image tasks is fully demonstrated, demonstrating the superiority of the method from multiple perspectives.

**Weaknesses:**

1.Table 2 compares the performance of SETOKIM method and some similar methods under the task of zero-shot text-to-image generation and editing under several data sets. The results show that the indicators of some data sets are not optimal, and there is still some gap with SOTA.

**Questions:**

1.In Table 2 that SETOKIM and some similar methods still have a gap in performance with SOTA under some data sets, please give some explanations.

---

### Meta-Review · Area_Chair_BDoz · 2024-12-24

**Metareview:**

This work purposes a novel dynamic Semantic-Equivalent Vision Tokenizer (SeTok)，using a dynamic cluster method to group visual features into semantic units, flexibly determining the number of tokens based on image complexity. The proposed SETOK-based MLLM (SETOKIM) shows superior performance on various text-image tasks. Although reviewers have raised some concerns about performance comparison, missing experimental results and so on. The authors have addressed these concerns well in the discusion period. Therefore, the AC recommends this paper as accept.

**Additional Comments On Reviewer Discussion:**

In the discussion period, the concerns of four reviewers' have been well addressed. Although reviewer #QXWj is absent in the discussion, I think his/her concern have been addressed by the authors' by carefully reading the comments and authors' responses.

---

### Decision · Program_Chairs · 2025-01-22

Accept (Poster)